# Effects of Peroxide Initiator on the Structure and Properties of Ultralow Loss Thermosetting Polyphenylene Oxide-Based Composite

**DOI:** 10.3390/polym14091752

**Published:** 2022-04-26

**Authors:** Xueyi Yu, Zeming Fang, Qianfa Liu, Dan Li, Yundong Meng, Cheng Luo, Ke Wang, Zhiyong Lin

**Affiliations:** 1School for System Design and Intelligent Manufacturing, Southern University of Science and Technology, Shenzhen 518055, China; yuxy2020@mail.sustech.edu.cn (X.Y.); 12031292@mail.sustech.edu.cn (Z.F.); 2College of Materials Science & Engineering, Huaqiao University, Xiamen 361021, China; 3National Engineering Research Center of Electronic Circuits Base Materials, Shengyi Technology Co., Ltd., Dongguan 523000, China; liuqf@syst.com.cn (Q.L.); lid@syst.com.cn (D.L.); mengyd@syst.com.cn (Y.M.); luoc@syst.com.cn (C.L.)

**Keywords:** copper clad, thermosetting PPO, dielectric properties, peroxide initiator

## Abstract

Although thermosetting polyphenylene oxide- (PPO) based composites with excellent dielectric properties have been widely accepted as superior resin matrices of high-performance copper clad laminate (CCL) for 5G network devices, there has been limited information regarding the composition–process–structure–property relationships of the systems. In this work, the effects of peroxide initiator concentration on the structure and dielectric properties of a free radical cured ultralow loss PPO/Triallyl isocyanate (TAIC) composite system were studied. As expected, the glass transition temperature (T_g_) and storage modulus increased with the advancing of crosslinking, whereas the dielectric loss showed an “abnormal” rise with the increase in crosslink density. Extensive studies were carried out by varying the initiator contents and characterizing the structure with spectroscopy, thermal analysis, and positron annihilation lifetime spectrum (PALS) techniques. The results show that the competition of polarity, crosslink density, free volume, and free TAIC are the key factors determining the dielectric properties of the composites.

## 1. Introduction

The rapid development of 5G communication technology has initiated a large-scale, worldwide effort to develop more sophisticated electronic systems requiring high-performance dielectric materials for printed circuit boards (PCBs). The main development directions of PCBs in the 5G era are a high layer count design, high frequency, and high levels of signal integrity, as well as high wiring density, which have incurred more stringent demands on the PCB substrate materials, i.e., copper clad laminates (CCL). In addition to lead-free compatibility and reliability requirements, a low dielectric constant (D_k_) and extremely low dielectric loss (D_f_) under high frequency are the most challenging material properties which have attracted great attention in both industry and academic institutions [1,2,3,4,5].

The D_k_ and D_f_ of traditional thermosetting CCL matrix materials, such as epoxy resin and cyanate ester (CE) resin, are intrinsically too high to meet the requirements of high-frequency applications [6,7]. Therefore, new polymer systems with low D_k_ and D_f_ are being developed for high-frequency and high-speed applications. Among the available polymers, polyphenylene oxide (PPO) has great potential since it has a highly rigid skeleton structure with aromatic rings, without hydrolysable groups or polar side groups. These structural characteristics endow PPO with excellent hydrolytic stability, low moisture absorption, high thermal stability, and excellent electrical properties in a wide temperature range [8,9,10,11]. 

For PCB applications, thermosetting PPO systems with excellent performance have been developed by capping the ends of PPO oligomers with reactive groups and curing with chemicals containing multiple functional groups, such as triallyl isocyanurate (TAIC), divinyl benzene, and 1,2-polybutadiene (PB), etc. [12,13,14,15,16,17,18], or copolymerizing with epoxy, cyanate ester, and benzoxazine, etc. [19,20,21,22,23,24,25,26,27,28,29]. Among the most significant developments, Peters et al. [30] invented a methyl methacrylate capped bi-functional PPO oligomer (M-PPE-M) which was later commercialized by SABIC under the tradename of NORYLTM SA9000 and studied several cured systems with t-butyl styrene, divinyl benzene, TAIC, dibrominostyrene, etc., as crosslinking agents [31,32,33]. Outstanding dielectric, thermal and mechanical properties, as well as good water resistance, were achieved simultaneously in these systems. For example, for the TAIC cured systems at 1GHz, the dielectric constant slightly changed between 2.5 and 2.57, whereas the dielectric loss was always as low as about 0.001 over a wide composition range (M-PPE-M/TAIC = 0% ~ 80%) [32]. Such dielectric properties were much better than that of traditional PCB substrate materials, such as epoxy-based (D_k_ ≈ 2.85, D_f_ ≈ 0.013 at 1GHz) and cyanate ester-based systems (D_k_ ≈ 2.73, D_f_ ≈ 0.0051 ~ 0.0117 at 1GHz) [32,34]. In the past decade, several PPO-based substrate materials with similar formulations have been successfully commercialized and have started to play key roles in 5G devices [12,13,14,15,16] and a lot of research has been carried out regarding the combination of PPO oligomers with different crosslink agents, copolymers, or reinforcement components (DVB, PB, Epoxy, CE, etc.) [17,18,19,20,21,22,23,24,25,26,27,28,29]. However, little work has been reported regarding the characterization of chemical structures and the relationships between crosslinked structures and the dielectric properties of thermosetting PPO-based ultralow dielectric loss systems.

The dielectric properties of polymers are related to their polarizability. The relationship between the molar polarizability *P*, the dielectric constant, and the molecular polarization α can be expressed in Equation [35]:$$P=\frac{\varepsilon_{r}-1}{\varepsilon_{r}+2}\times \frac{M}{\rho }=\frac{N_{A}\alpha }{3\varepsilon_{0}}$$
where P is the molar polarizability, $\varepsilon_{r}$ is the relative permittivity, $\varepsilon_{0}$ is the permittivity in vacuum, M is molecular weight of a repeat unit, ρ is density, α is polarizability, and $N_{A}$ is the Avogadro constant. 

Molecular polarizability refers to the proportionality constant for the formation of dipole under the influence of an electric field [35]. Based on the relation, all the molecular and structural parameters that enhance polarizability and density, such as the higher polarity of functional groups or atoms, higher molecular movability, inclusion of polar or conductive impurities, and less free volume, etc., will lead to a higher dielectric constant, and vice versa [35,36,37,38,39,40,41,42,43,44,45]. Dielectric loss is related to the energy dissipated due to the oscillating electric field. It is also related to the dielectric relaxation time τ, which is the time taken for the dipoles to return to their original position after the application of the electric field. When the rate of the electric field oscillate is much faster than the relaxation time, the polarization process in a molecule is unable to follow the rate of the change of the applied oscillating electric field and this results in energy absorption and dissipation as heat [35]. The molecular and structural parameters affecting polarizability usually have similar effects on dielectric loss. Synchronous changes of dielectric constant and dielectric loss have been observed in most studies.

For thermosetting polymer systems crosslinked through free radical mechanisms, the type and concentration of initiator are the critical factors determining the dielectric properties by influencing the crosslinking density and creating low-polarity functional groups. Wu et al. [46] studied the blends of 1,2-polybutadiene/styrene-butadiene-styrene triblock copolymer/ethylene-propylene-dicyclopentadiene (1,2-PB/SBS/EPDM) blends cured with two different organic peroxides, cumyl peroxide (DCP) and bis(1-(tert-butylperoxy)-1-methylethyl)-benzene (BIPb). The results showed that BIPb is more reactive and more effective in achieving high crosslink density and converting –C=C– double bonds with relatively high polarizability into –C-C– single bonds with low polarizability, which leads to low D_k_ and D_f_. Higher initiator concentration resulted in lower D_k_ and D_f_ regardless of initiator type through similar mechanisms, i.e., higher cross-link density that more effectively hindered the movement of polar groups, and further reduced the polarizability by replacing more –C=C– double bonds with new –C-C– bonds. In addition, the effects of the by-products from the decomposition of initiators on the dielectric properties cannot be ignored [47,48]. Walker [47] investigated the effects of initiator (DCP) content on the dielectric properties of crosslinked polyethylene (XLPE). The results showed that, with higher initial DCP concentration, more byproducts were produced which resulted in an increase in dielectric loss. 

Thermosetting PPO-based systems using TAIC as the crosslink agent have been widely used as the matrix of high-performance CCLs [13,14,15,16,49,50,51]. Usually, the PPO/TAIC ratio ranges from 1.0:1.0 to 4.0:1.0 and more typically from 1.2:1.0 to 2.0:1.0, which ensures excellent dielectric properties and good processability. A low PPO/TAIC ratio gives lower D_f_ [32]; however, the prepreg starts to become sticky at room temperature which causes processing issues. In this work, a series of PPO/TAIC-based composites with ultralow dielectric loss were prepared with a fixed ratio of PPO/TAIC at 1.33:1.0 and a low loss glass fabric as the reinforcement. The effects of initiator concentration on the chemical structure, thermal-mechanical and dielectric properties of a thermosetting PPO/TAIC-based composite system were studied. The dielectric constant of the precured prepreg and extensively cured laminate samples decreased with the higher concentration of the free radical initiator, whereas the dielectric loss of the samples exhibited significantly different or even opposite trends. The mechanisms were discussed based on the competition of multiple factors that affect the polarizability of thermosetting systems.

## 2. Materials and Methods

### 2.1. Materials

Methyl methacrylate terminated PPO resin (NORYL^TM^ SA9000) with Mn = 2300 and Tg = 160 °C was purchased from SABIC (Riyadh, Saudi Arabia). Triallyl isocyanurate (TAIC, 98%, AR) was supplied by Shanghai Aladdin Bio-Chem Technology Co., LTD (Shanghai, China). The free radical initiator, BIPb-96 (mixture of bis(tert-butyldioxyisopropyl) benzene (40%) and di(tert-butylperoxyisopropyl) benzene (60%), 96%) was purchased from Hunan Enpai New Material Company (Changsha, China). Their chemical structures are shown in Figure 1. The solvent, methyl ethyl ketone (MEK, 98%, AR), was purchased from Shanghai Aladdin Bio-Chem Technology Co., LTD. The L-glass fabrics (Style 2116) were provided by Asahi Kasei E-materials Corporation (Tokyo, Japan). All the solvents and reagents were used without further purification.

### 2.2. Sample Preparation

SA9000, TAIC, and MEK were mixed with the compositions shown in Table 1 and Table 2 in a 3 L beaker by stirring the mixture for 120 min in a fume hood at room temperature. Then BIPb-96 was added to the beaker and stirred for a further 60 min to obtain the varnish. The solid content of the vanish was set at 70% by weight which ensured sufficient dissolution and dispersion of components and maintained the appropriate viscosities for impregnation.

Pieces of glass fabrics were fully immersed in the varnish and passed through a pair of steel bars with preset gaps to make sure the glass fabrics were uniformly coated with specific amount of varnish. The coated glass fabrics were placed in a fume hood to dry for 24 h and then baked in an explosion-proof oven at 150 °C for 7 min to obtain the prepreg samples. Extensively cured laminate and cast samples were prepared by curing prepreg or resin powders taken from the prepreg samples, respectively. In addition, a P/T-0.00-L sample (without initiator) was air-dried at room temperature. The curing procedure is shown in Figure 1. 

### 2.3. Measurements

The solid-state nuclear magnetic resonance (SSNMR) spectra ware recorded on a Bruker III 400 M spectrometer (Rheinstetten, Germany) equipped with a solid-state high-resolution apparatus. The conventional cross-polarization/magic angle spinning (CP/MAS) method was used for high-resolution solid-state ^13^C measurements, and the spinning rate was 12 kHz.

Fourier transform infrared spectra were recorded on a PerkinElmer Fourier transform infrared reflection (FTIR) spectrometer Spotlight 400 N (Shelton, FL, USA)with a scanning step of 2 cm^−1^ and accumulation of 8 scans. Both the KBr tableting method and universal attenuated total reflectance accessory (UATR) were applied. For the KBr tableting method, the sample was mixed with KBr powder in a proportion of 1:100–1:200 and the total weight was controlled to 200 mg. For quantitative studies, the weight of samples and KBr powder were precisely weighed to 0.01 mg before mixing.

High-performance liquid chromatography (HPLC) was performed on a Waters e2695 (Milford, CT, USA) series liquid chromatograph system, which was equipped with a diode array detector (DAD) controlled by an Empower chromatographic workstation. A Sunfire C18 (150 cm × 4.6 cm, 5 μm) was adopted for the analysis. The mobile phase consisted of acetonitrile (chromatographically pure, 70%) and ultrapure water (30%) using an isocratic elution. The mobile phase was set at a flow rate of 1.0 mL/min and the column temperature was conditioned at 25 °C.

The Liquid-state nuclear magnetic resonance (NMR) spectra were measured on a Bruker Avance III 400 M with BBO probe (Rheinstetten, Germany). Samples were dissolved in deuterated chloroform (CDCl_3_) using tetramethylsilane (TMS) as the internal reference at room temperature.

Dynamic mechanical analysis (DMA) was performed in a tension mode using NETZSCH DMA 242E (Selb, Germany) from room temperature to 300 °C at a heating rate of 2 °C min^−1^ and a frequency of 1 Hz in air. The specimen dimensions were about 15 mm × 5 mm × 0.4 mm.

Melt viscosity (MV) was measured on a Brookfield CAP2000+H (Toronto, ON, Canada) at 171 °C and with spindle rotating speed 23 rpm. The minimum readings within 60 s were recorded as the MV.

Dielectric constant and dielectric loss were tested using an Agilent Technologies ENA series network analyzer (E5071C, Santa Clara, CA, USA) with the split-post dielectric-resonator (SPDR) at 10 GHz, 50% relative humidity, and room temperature. Before testing, each sample with thickness about 0.4 mm was dried in a 110 °C oven for 1 h and then cooled to room temperature in a desiccator for 30 min before testing.

Differential scanning calorimetry (DSC) curves were obtained with a NETZSCH DSC 214 (Selb, Germany)from room temperature to 300 °C under a nitrogen atmosphere with a heating rate of 2 °C·min^−1^.

Free volume was obtained by positron annihilation lifetime spectrum (PALS, DPLS3000, Anhui, China). There were two identical samples of the size at 1.5 mm × 10 mm × 10 mm in a 1.1 × 106 Bq ^22^Na positron source. And a 1.28 MeV γ-ray was emitted by the ^22^Na nucleus and the positron within a few picoseconds. The positron lifetime was determined by the time delay between the emission of the birth gamma (1.28 MeV) and an annihilation photon of 0.511 MeV. PALS were obtained using the fast–fast coincidence system. The time resolution was 220 ps and the channel width was 12.6 ps. Program PATFIT was used to process the data and get the lifetime spectra. The positron source components (392 ps/16.40%, 2.05 ns/0.70%) were subtracted before analyzing each spectrum. The variance of the fits was about 1.1.

## 3. Results and Discussion

### 3.1. Chemical Structure of Cured Matrix

The SSNMR spectra of cured samples were shown in Figure 2. The peak positions were consistent for the two samples, indicating the same curing mechanisms and no changes in chemical structure were induced by the varied concentration of initiators. The signals at 126.0 ppm and 136.5 ppm coming from –C=C– of SA9000 [52] were not detected, showing that all the –C=C– of SA9000 had participated in the reaction.

The FTIR spectra of cured P/T-0.25 and P/T-1.50 together with that of SA9000 and TAIC were shown in Figure 3. The FTIR spectra of P/T-0.25 and P/T-1.50 were identical in terms of the number and position of the peaks, which agreed with SSNMR results that there are no differences in chemical structure caused by initiator concentrations. 

The existence of the absorption peaks at 1645 and 3086 cm^−1^ suggested that some–C = C– groups from TAIC did not participate in the crosslinking reaction [52]. To quantify the amount of unreacted –C = C– groups from TAIC, the peak at 3086 cm^−1^ was employed for the calculation using Lambert–Beer law [53] with the following steps: 

The powder from P/T-0.00-L was weighed out in five different amounts and thoroughly mixed with the appropriate amount of KBr powder to keep the total mass of the mixture at ~200 ± 0.5 mg. The mixture was pressed into a disc for the FTIR scanning. The obtained spectra were shown in Figure 4 and the calculated mole numbers of –C = C–(n_–C=C–_ ) in TAIC in the samples were listed in Table 3. The peak areas were plotted against the n_–C=C–_ in Figure 5 and a linear regression equation y = 74.79x was obtained with the R^2^ = 0.975. By calculating the peak areas obtained from the FTIR spectra of cured P/T-0.25 and P/T-1.50 (Figure 3b), the residual ratio of the –C=C– of TAIC in these two samples was 23.33% and 17.5%, respectively, revealing that more –C=C– in TAIC participated in the reaction as the mass fraction of the initiator increased.

### 3.2. Unreacted TAIC in Laminates

Matsumoto et al. [54] studied the curing reaction of TAIC and pointed out that the unsaturated bonds of TAIC are difficult to be fully consumed because TAIC chains have bulkier isocyanurate side groups than the chains derived from the polymerizations of other multi-allyl compounds, and rigid TAIC chains lead to a reduced interaction between polar isocyanuric rings belonging to both the polymer chain and TAIC monomer. These dissociated –C=C– bonds either came from the partially reacted TAIC molecules or from the unreacted free TAIC molecules in the system (Figure 2). To identify and quantify the soluble components of the cured samples, a certain amount of cured P/T-0.25 and P/T-1.50 samples were crushed and extracted for 12 h in boiled MEK and the extracts were studied by HPLC, ^1^H-NMR, and ^13^C-NMR.

The standard curve for quantitative HPLC analysis was established by measuring several TAIC/MEK solutions with a given TAIC concentration, as shown in Figure 6a,b. Based on the linear regression equation obtained and the chromatograms of the extracts (Figure 6c), it was found that there was 276.72 mg of TAIC in the extract of P/T-0.25, which meant that 14.74% of the TAIC did not take part in the reaction and it occupied 6.24% of the total mass of the sample. Furthermore, in P/T-1.50, these values were 7.49 mg, 0.48%, and 0.20% (Table 4).

Similar phenomena were observed with NMR measurements. ^1^H-NMR and ^13^C-NMR spectra of the extracts were shown in Figure 7a,c. When Figure 7a,c were enlarged in equal proportions (Figure 7b,d), similar phenomena as the HPLC could be obtained. There was 286.29 mg of TAIC in the extract of P/T-0.25, but not in P/T-1.50. It was found that 15.26% of the TAIC in P/T-0.25 did not participate in the reaction, which was 6.46% of the total mass of the sample (Table 3).

### 3.3. Thermal-Mechanical Properties and Free Volume

Dynamic mechanical analysis (DMA) was used to characterize the T_g_ and the crosslink density ($v_{e}$) of the samples. The storage modulus ($E^{\prime }$) curves and the tanδ curves as a function of temperature for samples with different initiator mass fractions were shown in Figure 8a,b. The peak temperature of the tanδ was denoted as T_g_. The storage modulus ($E^{\prime }$) of DMA in the rubber plateau area (40 °C above T_g_) was used to calculate the $v_{e}$ based on the following formula [55,56,57]:$$v_{e}=\frac{E^{\prime }}{3RT}$$
where $E^{\prime }$ was the storage modulus of the thermoset in the rubbery plateau region at Tg + 4 °C, R was the gas constant, and T was the absolute temperature at Tg + 40 °C.

The Tg and $v_{e}$ of samples were shown in Table 5. As expected, with more initiator added into the system, the crosslinking reaction advanced more extensively, which resulted in a higher Tg and higher crosslink density. 

PALS was used to measure the free volume of different samples in the system. As shown in Figure 9 and Table 6, the free volume became larger with the increased initiator content.

### 3.4. Dielectric Properties

As shown in Figure 10a, at 10 GHz with a resin contents of about 52% the cured PPO/TAIC/L-glass composites exhibited dielectric constants from 3.24 to 3.10 and ultralow dissipation factors from 0.0023 to 0.0031, respectively. Before curing, the dielectric constants of prepreg samples decreased significantly with higher initiator concentration, while the dielectric loss stayed almost the same. After curing, the dielectric constant of the laminates kept the same trend as that of prepreg but moved to higher levels compared to the corresponding prepreg samples. The dielectric loss, however, showed an ascending trend at the same time. This trend was further confirmed with the resin cast samples (Figure 10b), indicating that the variation of the dielectric properties of the composites was mainly caused by the intrinsic variation of the resin matrix.

For thermosetting polymer systems, it had been commonly agreed that, with the increase in crosslinking density, the orientation of the groups is progressively hindered and usually results in a lower dielectric constant and dielectric loss [43,44,45]. In the current PPO/TAIC system, however, it was obvious that more polarization and relaxation mechanisms were involved, and the significance of each mechanism varied depending on the advancement of curing, environment of the functional groups, and crosslinking structures.

When PPO and TAIC were mixed together with a ratio of 4:3, the TAIC could be considered as a reactive plasticizer before curing since the two components were fully miscible at this composition [58]. It has been revealed that plasticizers in polymers can reduce intermolecular interaction, enhance the segmental mobility of polymer chains, and promote the dipole moments, thus increasing the dielectric constant [59,60]. Such mechanisms may play important roles in the current system.

While baked at 150 °C, some polymerization reactions were initiated which resulted in an increase in melt viscosity (Table 7). The DSC scan of P/T-0.25-L prepreg in Figure 11 showed that the curing reaction started at about 120 °C and had dual exothermic peaks at 160 °C and 180 °C, respectively, whereas that of pure TAIC started at 100 °C and reached an exothermic peak at 155 °C. The results revealed that multiple reaction processes are involved in the curing of PPO/TAIC systems. It is most likely that the peak at 160 °C can be mainly attributed to the curing of pure TAIC, and the reaction between PPO and TAIC occurred later at higher temperature. FTIR spectra in Figure 12 and the calculated results in Table 8 confirmed that, with the increase in initiator content in original compositions, more TAIC participated in homopolymerization and less plasticizer was left in the matrix. Thus, the plasticization effect was weakened, which explained the decrease in the dielectric constant of prepreg samples with higher initiator concentration. On the other hand, however, the dielectric loss at the prepreg stage did not change much with the variation of initiator content and the level of TAIC polymerization, which can be explained by the fact that the TAIC monomers have a highly symmetric structure and poly(TAIC) has a very low dielectric loss [32]. 

When the samples were cured at elevated temperatures, several changes took place in the system. First, with the advancement of the curing reaction, more and more TAIC molecules reacted with SA9000 and either generated crosslink networks by bridging SA9000 molecules or formed bulky pendant groups by grafting onto SA9000 chains. Both factors caused decreased dipolar mobility, contributed to lower polarizability, and drove the dielectric constants to a lower level. At the same time, the lessening of the plasticizing effects of TAIC were also in favor of the reduction of the dielectric constant. On the other hand, however, there were several new factors generated during the curing which may have promoted an increase in dielectric constants. PPO and TAIC molecules are highly symmetrical, and the polarity was very low. When PPO and TAIC were copolymerized, the structural symmetry of both components was diminished and the dipole movements were enhanced, which resulted in significantly higher molecular and group polarity. Moreover, short TAIC branches can weaken the interaction force between polymer chains and increase the movability of polymer chains. Furthermore, chain transfer products from free radicals may also contribute to the overall polarizability of a system as polar impurities. For example, the primary radicals from the initiator may have been terminated by taking atoms from the other reactant and producing hydroxyl groups. Although the presence of alcoholic hydroxyl groups was not detected by ATR-FTIR or SSNMR [52], such byproducts from chain transfer reactions were inevitable and contribute to an increase in dielectric properties.

Therefore, the overall trends of dielectric constant and dielectric loss were the superposition of these competing factors. It was evident that the factors increasing polarizability prevailed after curing compared to that of the corresponding prepreg samples, which resulted in an increase in dielectric constant and loss. Moreover, the polarizabilities were enhanced with the higher concentration of the initiator which led to the significant ascending of dielectric loss, whereas increases in the dielectric constant were partially offset by the increased free volume and reduced plasticizing effect at higher initiator concentration (Figure 9 and Table 6).

## 4. Conclusions

The PPO/TAIC-based polymer composites exhibited excellent dielectric properties which are suitable for high-frequency applications. In this work, the composition–process –structure–property relationship of PPO/TAIC-based polymer composites with ultralow dielectric loss was studied. Special attention was focused on the effects of peroxide initiator concentration upon the dielectric properties. For prepreg samples, the plasticizing effect of TAIC played a key role which caused a decrease in the dielectric constant with the initiator content, while it did not affect dielectric loss much. When the samples were cured at high temperatures, more mechanisms affecting polarizability developed with the advancement of the reaction, including some factors that contributed to lower polarizability, such as increased crosslink density, free volume and quantities of bulky pendant groups, and reduced plasticizing effect, etc., as well as some factors that promoted polarizability such as enhanced polarity, diminished molecular symmetry, branching, and hydroxyl impurities, etc. As a result of competition, the latter mechanisms dominated the trends and both the dielectric constant and dielectric loss of the cured samples increased significantly with higher initiator concentration compared to the corresponding prepreg samples. 

## Data Availability

The data presented in this study are available on request from the corresponding author.

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
