# Peer review of "Effects of Peroxide Initiator on the Structure and Properties of Ultralow Loss Thermosetting Polyphenylene Oxide-Based Composite"

_polymers, 2022, doi:10.3390/polym14091752_

Round 1
Reviewer 1 Report
The manuscript investigated the influence of crosslinking density on the performance of thermosetting polymer by tuning the initiator composition. The results showed that the thermodynamic, thermal-mechanical, and dielectric properties of polymer composite varied considerably with the change of crosslinking density. Various spectrums were conducted to study the underlying mechanism. Overall, the experimental results are clear, and the discussion is reasonable. Some issues should be addressed to further improve the manuscript before publication.
- What played the dielectric constant and dielectric loss key in the polymer composite?
- The change of PPO composites' dielectric properties suggested varying slightly by adjusting initiator concentration. What are the results of reported references? A detailed comparison should be conducted for clarity. More discussions are also helpful to understand the mechanism.
- The authors explained the crosslinking degree of polymer composite by calibrating the spectrum; how about the resolution? This method always functioned in a quantitive way with low resolution. Any other ways to determine the crosslinking degree of resultant polymers?
- The figures in the manuscript were very messy, with different image sizes, font sizes, and styles. The quality and styles of all the figures should be modified to be consistent.
- Some details are missing in the manuscript, especially the equations, and the authors should carefully check the draft.
Reviewer 2 Report
By the topic, the article titled “Effects of peroxide initiator on the structure and properties of ultralow loss thermosetting polyphenylene oxide based composite” appears as extremely interesting. A preliminary reading of the text confirms this impression. However, a more quiet and profound study of the article makes emerge a series of unclear concerns that must be better explained by the authors as to permit to this reviewer recommend the acceptance of the paper.
There are some minor and major concerns.
- Minor: In the introduction section the equation quoted as [33] is missing in the text. Please, include it.
- Minor: Equally, in lines 69 and ss., the signs for permissivity, polarizability, density, and Avogadro number are missing too. Please, include them in the text.
- Major: Please, explain having used only the SA9000/TAIC/MEC 4/3/3 ratio and not others.
- Minor: provide the number of scans used in FTIR experiments.
- Major: Since other peaks may be expected in DMA above 300ºC, this reviewer wonders why the authors have limited the study to 300ºC when very probably other higher thermal transitions will occur providing more valuable information to the system performance. (Note that this is what happens to the PPO in isolation).
But the main (and very MAJOR) concern is related to the absence of the P/T 0.00 sample in all the figures when this one (the only one with the absence of an initiator) may be included for mere comparative reasons (or convincing rebuttal). This reviewer wonders why the authors have avoided including this one.
In the same way, with the exception of the DMA study that considers all the samples except the P/T 0.00, all the other techniques (besides avoiding the P/T 0.00 one) have been performed only on a set of very limited samples without explaining the criteria followed in such practice, making difficult to ascertain the final purpose. Just as mere examples, below these lines some indications are given:
- In Figure 2 the P/T-0.25 and P/T-1.50 samples are compared. Why not any other?
- In figure 3. Just the same. Why not the reference sample without an initiator? It is true that this one has been included in Figure 4b but under a different "y" scale. Note that the patterns shown may be very similar to the ones in Figure 3 if represented under the same scale.
- Similarly for Figures 6 and 7.
- However, Figure 8 (related to the DMA study) included all the samples except the P/T 0.00. This reviewer wonders about the criteria for including here almost all the spectra since the evolution found is just the expected (even for a layman), and has failed in including those in the other figures where the effect may not be so evident. Provide a robust rebuttal.
- This reviewer wonders why figure 9 chooses now the sample P/T-1.25 instead of the one chosen for the other techniques (P/T-1.50). Any scientific reason for it?
- Other similar concerns may be made for the rest of the figures.
- This reviewer wonders why the authors have merely used four samples (the P/T-0.25; The P/T-0.75, the P/T-1.25; and the P/T-1.50) when the experimental plan in Table 1 consists of 7 different samples.
In summary, the authors present partial data for four samples, avoiding three of the experimental plan, giving the impression to the reader of having chosen the ones conducting a seemingly pre-established conclusion. The authors must answer all these queries convincingly before recommending the publication of the article, despite (and because of) its great interest.
Reviewer 3 Report
In this work, the effects of peroxide initiator concentration on the structure and dielectric properties of a free radical cured ultralow loss PPO/Triallyl isocyanate (TAIC) composite system were studied. The paper is interesting and could be published after revision.
-How the authors decided that the free radical initiator BIPb-96 is well suitable for the polymerization ?
- How the author have found that these conditions are well suitable for preparation of the prepreg samples: in an explosion proof oven at 150°C for 7 minutes ?
-The authors should describe clearly advantages and disadvantages of the composites as compared with those of other materials used in this field of applications.
-Could properties of the new composites be compared with other commercially available products used in this field ?
-The advantages and disadvantages should be well described in conclusions.
Round 2
Reviewer 2 Report
See the attached file.

Reviewer 3 Report
If editor and other reviewers agree that the paper is suitable for “Polymers” I would recommend the paper for publication after the revision.
Author Response
Thank you for your review and confirmation.